# Travelling from Perspective of Persons with Disability: Results of an International Survey

**DOI:** 10.3390/ijerph191710575

**Published:** 2022-08-25

**Authors:** Urszula Załuska, Dorota Kwiatkowska-Ciotucha, Alicja Grześkowiak

**Affiliations:** 1Department of Logistics, Wroclaw University of Economics and Business, 53-345 Wrocław, Poland; 2Department of Econometrics and Operational Research, Wroclaw University of Economics and Business, 53-345 Wrocław, Poland

**Keywords:** people with sensory disabilities, inclusive tourism, statistical analysis of survey results

## Abstract

Full inclusion of people with disabilities means their full participation in community life and the same opportunities to work and spend their free time that other members of the community have. This also applies to travel and tourism. Offers available to people with various types of disabilities are seldom adapted to their needs. They face numerous barriers and obstacles when travelling or at their destination. The article presents selected results from an international comparative study concerning travel of people with sensory disabilities. The study was carried out in the first quarter of 2022 using the PAPI method on a group of 131 respondents from Poland, Greece, Cyprus and Portugal. To analyse the results, we relied on statistical inference using an independent two-sample *t*-test and one-way analysis of variance. Tests of the equality of two means were preceded by Levene’s test for homogeneity of variances. According to the study, people with sensory disabilities can see many barriers to travel that pose a significant constraint on their activity. These barriers vary depending on the type of disability, gender or the country of origin of the respondent, but the list of indications often includes the need to train service staff in the specific needs of people with different types of disabilities. Taking into account development opportunities that people with disabilities create for the tourism industry, including people with sensory disabilities who are frequently overlooked, it is worth considering measures aimed at improving knowledge and skills in this area in the future.

## 1. Introduction

According to WHO reports, more than one billion people worldwide, that is, approximately 15% of the population, are affected by some kind of disability. This number is increasing dramatically due to ageing populations and the spread of chronic diseases [1]. Although these people are guaranteed the same rights as other community members, they often face barriers in education, employment, transport, etc., that significantly limit their activity. The Convention on the Rights of Persons with Disabilities [2], adopted in 2006 by the United Nations and ratified by all European Union countries, ensures full inclusion of people from this group, but even the best legal provisions without any specific actions remain only declarations on paper.

The full inclusion of people with disabilities means increasing their activity in public spaces, including improving the accessibility of various leisure activities. Accessible tourism is becoming increasingly popular, with more and more tourist destinations, hotels or restaurants claiming that they are open to people with disabilities. Unfortunately, this openness is often apparent, and the issue of adapting hotels or tourist places to the real needs of people with disabilities requires a change in attitudes towards disability and a better understanding of the expectations of this target group [3,4,5,6,7,8,9]. Openness is often associated only with accessible architecture and investments in lifts, ramps or other facilities for people who have problems with mobility. It is frequently forgotten that a disability is a diversity of needs and expectations and that each type of disability requires an individual approach. It means that people with disabilities are still a new target group for the tourism industry. This target group, on the one hand, often requires a non-standard approach, but on the other hand, because of its size, it creates enormous growth opportunities for the industry. The period of the COVID-19 pandemic showed that the use of modern technology could effectively increase accessibility for people with cognitive or sensory disabilities. Smartphones with audio descriptions and the possibility of translating texts directly into sign languages or online communication tools are just a few examples of ICT applications for increasing accessibility.

In the literature, attention is paid to the relationship between inclusive tourism and social and economic development [10], and the industry is believed to have the potential to provide additional social and economic benefits [11]. What is more, Liasidou et al. [12] emphasise that the issues can be analysed in a broader spectrum and concern other groups of people with special needs (e.g., elderly people, pregnant women or people travelling with young children). The research on accessible tourism concerns different aspects connected with travelling, some of which are related to transport [13,14,15,16,17]. Other research concerns the accessibility of accommodation [3,18,19,20,21] and different tourist attractions [4,5,22,23,24].

Increasing accessibility requires knowledge of the diverse needs that people with disabilities have and the ability to put this knowledge into practice. Hence, the importance of field research focused on characterising the needs and expectations of people with specific disabilities and the training of service staff. The source of funding for such primary research and training activities can be external, e.g., as part of projects financed by the European Social Fund or the Erasmus+ Programme [25,26,27]. In the case of projects funded by these sources, the diagnosis of needs and support for people with disabilities are the major focus.

An example of a project that directly addresses the specific needs of people with sensory disabilities is the Erasmus+ strategic partnership project *Time4Alternative creativity in remote space* (Programme Erasmus+, No 2020-1-PL01-KA227-ADU-095575). As part of the project, we planned primary research on broadly understood travel in a group of people with sensory disabilities. The analysis of selected results is presented in the article. 

For the purpose of this article, we formulated two research questions:

The first research question: What barriers to travel are particularly troublesome for people with sensory disabilities? In other words, what difficulties in travelling do people with sensory disabilities mainly point out? 

The second research question: Do respondents’ characteristics such as gender, nature of disability (people with hearing or vision problems) or country of residence differentiate the perception of travel barriers? In other words, do these characteristics necessitate the preparation of a dedicated offer tailored to specific needs?

The answers to both research questions are of great importance for making appropriate management decisions in the tourism industry. They can be treated as a source of information, about the actual needs and expectations of the developing but still little known to the industry target group.

## 2. Materials and Methods

Analyses are based on the results of the quantitative research carried out in the fourth quarter of 2021 in four European countries: Cyprus, Greece, Poland and Portugal. The study was conducted using the Paper and Pen Personal Interview (PAPI) method as part of the international *Time 4 Alternative Creativity in remote space* project. The sample was purposive, and the respondents were people with sensory disabilities (deaf, hard of hearing, blind, visually impaired, deaf-blind) or multiple disabilities (including sensory disabilities), mainly those with experience of travel. The aim of the survey was to gather opinions concerning selected aspects of travel, including destinations, meeting travel needs, presence of accompanying persons, barriers encountered and factors facilitating travel, criteria and accommodation facilities. The survey was conducted by project team members from the four selected countries. The respondents took part in the survey voluntarily. The opportunity to opt-out was guaranteed anywhere in the survey. The respondents were also guaranteed the confidentiality of their responses in accordance with the applicable data protection law (GDPR). The respondents answered questions about their experiences connected with both international and domestic travel. The survey questionnaires were translated into the national languages of all participating countries. As far as Cyprus and Greece were concerned, we used the same translation. The survey was carried out face to face by trained project team members in the respondents’ countries of residence. As for deaf respondents using sign language, a sign language interpreter specific to their country of residence was involved in completing the survey. The survey was prepared in a form accessible to visually impaired or blind people (audio description). The questionnaire was completed by the interviewer on the basis of the respondent’s statements, who also ultimately accepted the interview transcripts.

During the study, we collected 131 questionnaires—129 were complete, and the 2 remaining interviews were conducted with people who had no travel experience but were able to express their opinions on the barriers and needs of people with sensory disabilities. The characteristics of the respondents in terms of the specific features included in the survey questionnaire are shown in Table 1.

The distribution of characteristics presented in Table 1 is due to the purposeful selection of the sample and is not representative (impossible to compare to the population due to the lack of relevant population data). In the study group, 121 people reported sensory disabilities as the predominant type of disability. For these respondents, detailed analyses were carried out on the differences between people declaring vision (33 people) and hearing problems (88 people) as the main factor. The respondents were characterised by a different status in the labour market—the majority were employed (78 people), while others were retired (25 people) or unemployed (23 people).

In the article, we relied on statistical inference using an independent two-sample *t*-test and one-way analysis of variance. Tests of the equality of two means were preceded by Levene’s test for homogeneity of variances [28]. When heterogeneity of variance was found, an alternative to the classical approach, the Welch *t*-test statistic, was applied [29]. When ANOVA results showed significant differences, post hoc Tukey’s HSD tests for multiple comparisons [30] were carried out to identify the pairs characterised by different means. Calculations were performed using SPSS software (Armonk, NY, USA, IBM Corp).

## 3. Results

At the beginning of the survey, the respondents were asked to indicate the purpose of their travel. Most of them stated it was touristic (65.7%), followed by leisure (22.9%), other (11.5%) and business (8.4%). It should be noted that tourism was understood far more broadly than leisure, which was clearly visible in the questionnaire: “not only leisure, but also e.g., sightseeing, visiting tourist attractions, participating in the activities of local cultural and creative institutions”. 

The hierarchy of travel purposes for people with sensory disabilities is the same, with deaf people more often declaring tourism (66.7%) than blind ones (57.7%), while leisure appeared more frequently in the answers given by blind (34.6%) rather than deaf people (22.2%).

Most people with disabilities do not travel alone when taking trips with at least one overnight stay (Table 2). This way of travelling was reported by only 14.5% of respondents. In most cases, the respondents travel with friends (35.1%), a spouse/partner (32.8%) and family members (29.8%). A small percentage of people choose a travel assistant (2.3%). Some people indicated more than one answer.

The results concerning travel companions vary depending on gender. Women are far more likely to travel with friends (43.3%) than men (27.1%), while men are more likely to travel alone (20.3%) than women (10.4%). None of the women declared travelling accompanied by strangers or travel assistants. 

The second characteristic that differentiates travel companionship is the type of sensory disabilities. Deaf people mainly declared travelling with friends (43.2%), family (34.6%) and a spouse/partner (25.9%), while blind people declared travelling with a spouse/partner (34.6%), friends (30.8%) and alone (26.9%).

People with disabilities are particularly subject to specific barriers and obstacles when travelling, and only 6.1% of respondents declare that they do not encounter any of them.

Encountered difficulties might be of a different nature—financial, informational, health, architectural or related to lack of appropriate support (Table 3). The two most important issues refer to *Insufficient financial resources* (50.4%) and *Lack of adequate information about the attractions* (47.3%). For more than 1/3 of respondents, problems concern No *attractions or facilities dedicated to people with disabilities available at travel agencies* (36.6%) and *Lack of well-trained staff* (33.6%), followed by health condition, language skills, lack of a travel companion and architectural and transport barriers. Interestingly, people with disabilities do not face the problem of *Lack of support/acceptance of trips from family or friends* (3.8%).

The hierarchy of obstacles looks differently for women and men. According to female respondents, the five most important issues include: *Lack of adequate information about attractions, Insufficient financial resources, Lack of attractions, Lack of well-trained staff and Health state/disability*, while according to men, these are: *Insufficient financial resources, Barriers in the means of transport, Lack of attractions, Lack of adequate information about attractions and Lack of well-trained staff*. Most of the categories are repeated in a slightly different order, and the main difference is indicating *Health state/disability* by women and *Barriers in the means of transport* by men being among the most important ones.

Some differences in the hierarchy of obstacles can also be observed among people with different types of sensory disabilities. Although the top five obstacles are the same, their ranking is somewhat different. According to deaf people, the order is as follows: *Insufficient financial resources, Lack of adequate information about attractions, Lack of attractions, Lack of well-trained staff and Lack of knowledge of foreign languages*. Blind people, on the other hand, think that the biggest problems concern: *Insufficient financial resources, Lack of well-trained staff, Lack of adequate information about attractions, Lack of attractions and Lack of knowledge of foreign languages*. 

Table 4 shows the most important obstacles indicated by respondents in each country. *Insufficient financial resources* are the biggest problem among Greeks and Poles, while for Cypriots and the Portuguese, it is *Lack of adequate information about attractions*. The most important barriers were indicated in a similar way, although some country-specific elements have appeared. Only Polish respondents consider *Architectural barriers in accommodation* as a very important obstacle, while *Lack of travel companion* is a very important problem only for the respondents from Portugal.

As part of inclusive tourism, it is necessary to undertake measures that enable people with disabilities to take part in various activities. An evaluation of the importance of factors that could facilitate travel is presented in Table 5. The respondents attribute the greatest importance to *Offers from travel agencies dedicated to people with disabilities* (58.8%), which indicates that there is an insufficient supply of such services. Financial issues are very important, both in terms of better personal finances (51.1%) and *Co-financing of trips by public institutions/non-governmental organisations* (45.8%). Another issue is adequate information both on adjusting the attractions to the needs of people with disabilities (49.6%) and connected with tourist attraction websites (38.2%). The human factor was also frequently indicated—well-trained staff (45%) and the availability of an assistant for people with disabilities (38.9%). 

Female and male respondents indicated similar factors as the most important, and there was a strong convergence of opinions with the general results. The respondents from different countries created a hierarchy of factors in a similar way. Table 6 shows the most frequently indicated aspects. For Greeks and Poles, the most important aspect is the dedicated offer, whereas for Cypriots and the Portuguese, the possibility of receiving co-financing. As key factors, the respondents from all countries also indicated more financial resources (own), availability of information about attractions for people with disabilities and well-trained staff.

Factors indicated by people with sensory disabilities as the most significant are similar. According to deaf people, the most important ones include: *Offers from travel agencies dedicated to people with disabilities* (56.8%), *More own financial resources* (48.1%) and *Information about attractions for PwD* (48.1%). Blind people most often indicated: *Offers from travel agencies dedicated to people with disabilities* (61.5%), *More own financial resources* (57.7%) and *Information about attractions for people with disabilities* (57.7%).

Travel satisfies different needs, and its quality depends on a number of factors that may be specific to people with disabilities. The respondents assessed various aspects on a four-point scale, and their average evaluations are presented in Table 7. According to people with disabilities, the most important is the *Accessibility of facilities for people with disabilities* (average: 3.16). This was followed by *Tasting regional dishes and beverages* (average: 3.10), *Visiting monuments, works of art, theatres, etc.* (average: 3.00) and *Experiencing an adventure* (average: 2.79). The least popular aspects included *Visiting family and friends* (average: 2.58) and *Visiting craftsmen and learning about local handicrafts* (average: 2.28).

The significance of aspects was also evaluated in terms of differences by gender, type of sensory disabilities and country of the respondent. The test of the equality of means carried out for gender showed that opinions expressed by women and men differed in only one aspect—visiting monuments, works of art, theatres, etc. (t = 2.541, *p* = 0.014), where the mean of responses given by female respondents (3.23) was definitely higher than the mean of responses given by men (2.75). 

Significantly greater differentiation was found in the responses given by people with different types of sensory disabilities (Table 8). At the level of significance of 0.05, it can be concluded that a different evaluation applies to: *Tasting regional dishes and beverages* (t = 3.174, *p* = 0.002), *Experiencing an adventure* (t = 2.695, *p* = 0.008), *Improving health and well-being* (t = 3.474, *p* = 0.001), *Rest and relaxation* (t = 3.025, *p* = 0.003) and Visiting *craftsmen and learning about local handicrafts* (t = 2.513, *p* = 0.014). In each case, the evaluation of significance indicated by blind people was higher than the one indicated by deaf people.

One-way ANOVA shows that the importance of many aspects is evaluated differently by respondents from different countries (Table 9).

Only the accessibility of facilities for people with disabilities was evaluated in the same way, bearing in mind that, according to the respondents, this is the most important aspect. A post-hoc test (Tukey’s HSD) was carried out to determine in detail where the differences were. *Tasting regional dishes and beverages* obtained definitely higher evaluations from the respondents from Poland than from Greece (*p* = 0.006). Visiting *monuments, works of art, theatres* is more appreciated by the respondents from Cyprus than those from Greece (*p* = 0.000) and Portugal (*p* = 0.000). Poles also value this aspect more than Greeks (*p* = 0.009) and the Portuguese (*p* = 0.007). The respondents from Poland evaluated *Experiencing an adventure* in a more positive way than the respondents from other countries (Greece *p* = 0.008, Portugal *p* = 0.000, Cyprus *p* = 0.012). *Meeting new people* is more important for people from Greece than for those from Portugal (*p* = 0.023). Opinions on *Improving health and well-being* do not differ significantly only in the case of Greece and Cyprus—in the remaining cases, we found significant differences, with the highest value noted in Poland (average: 3.45), followed by Greece (average: 2.38), Cyprus (average 2.79) and Portugal (average: 1.54). The aspect of *Rest and relaxation* is more important in Poland than in Portugal (*p* = 0.000). The respondents from Poland value *Visiting family and friends* more than the respondents from other countries (Greece *p* = 0.000, Portugal *p* = 0.000, Cyprus *p* = 0.040), with Cypriots appreciating this aspect more than Greeks (*p* = 0.045). The Portuguese are less interested in *Visiting craftsmen and learning about local handicrafts* than Poles (*p* = 0.006) and Cypriots (*p* = 0.000). This aspect is more appreciated by Cypriots than Greeks (*p* = 0.000).

Particularly important for people with disabilities when travelling might be the place of accommodation. Criteria related to this were evaluated on a four-point scale, and the average results are presented in Table 10. The three most important issues concern *Location/good public transport connection to the city centre* (average: 3.60), *Additional services included in the price* (average: 3.50) and *Price* (average: 3.46).

The significance of the criteria was also evaluated for differences by gender, type of sensory disability and country of the respondent. The test of the equality of means performed for gender (Table 11) showed that the opinions expressed by women and men differed only for *Additional services included in the price* (t = −2.105, *p* = 0.037). This aspect was valued as significantly higher by men than women (average: 3.61 and 3.36, respectively). 

As for the type of sensory disability, significant differences were found for two aspects: *Possibility of cancelling the reservation free-of-charge* (t = 2.586, *p* = 0.011) and *Website dedicated to people with visual impairment* (t = 6.735, *p* = 0.000). Both of them are more valued by blind people. The second aspect is clearly relevant to people with visual impairment; hence this test result was expected.

One-way ANOVA shows that the importance of four aspects related to accommodation is evaluated differently by respondents from different countries (Table 12): *Price, Standard (quality of service), Possibility of cancelling the reservation free-of-charge* and *Website dedicated to people with visual impairment*. 

In order to determine in detail where significant differences occurred, we carried out a post-hoc test (Tukey’s HSD). The respondents from Cyprus evaluated the importance of price lower than the respondents from Portugal (*p* = 0.014) and Poland (*p* = 0.003). For Poles, the standard (quality of service) plays a greater role than for Greeks (*p* = 0.001) and Cypriots (*p* = 0.046). The possibility of cancelling the reservation free-of-charge is valued higher by Poles than by Cypriots (*p* = 0.017). A website dedicated to people with visual impairment is more important to Poles and Cypriots than to Portuguese (*p* = 0.002) and Greeks (*p* = 0.000).

## 4. Discussion

An international study conducted by our team in a group of people with disabilities shows that various barriers have a significant impact on limiting their travel activity. The findings are in line with those obtained by other research teams in the area of public transport accessibility or tourism [16,31]. Different types of disabilities mean different types of barriers, which result from different needs. However, some barriers stay the same regardless of the type of disability, for example, those related to poor awareness of the needs or lack of knowledge among service staff in the area of disability. Training in the specific needs of people with various types of disabilities, in the way of life when dealing with people with disabilities or in the possibilities of increasing openness provided by the development of modern technologies, is becoming a necessity if we think of this group as a development opportunity for the tourism industry [32,33]. Increasing openness to otherness means, first of all, increasing the level of knowledge about disabilities in society. Ignorance and lack of knowledge cause reluctance and passivity, whereas making people aware of the diversity of needs increases their empathy and openness to others.

The study aimed at evaluating the growth potential of the tourism industry shows that accessible tourism is a very important part of the market with high growth potential [34], also due to the fact that it can generate significant revenues in the future for major tourism providers. It is, therefore, worth preparing to serve this new and still little-known target group. It is also a good idea to think about modern technologies [35,36,37,38,39], the proper use of which opens up almost unlimited possibilities of perception by people with, for example, sensory disabilities.

The complexity of disability as a physical and social phenomenon requires in-depth research in the field of accessible tourism [40]. However, as the results of our research have shown, it is not only the type of disability that is a distinguishing feature in the area of perceived barriers to travel; the country of residence of a potential tourist with a disability is also such a characteristic. Just as able-bodied people from different countries perceive tourist attractions in a different way [41], people with disabilities from Cyprus, Portugal or Poland have a different hierarchy of needs and different perceptions of barriers to their activity. This is mainly due to cultural differences [42], which are visible in most areas of social functioning. For the tourism industry, it means that the specific characteristics of the countries of residence of potential tourists with a disability must also be taken into account when preparing the offer.

## 5. Conclusions/Future Research

Preparing the tourism industry to provide services to people with disabilities, or more broadly, people with special needs, is very important from two points of view. The social aspect allowing for the full inclusion of this target group in the community is crucial. However, the economic issue is important as well, mainly for the tourism industry—people with disabilities belong to a growing target group but are still little recognised by this sector. What is important for the success of both aspects is primary research that allows the voice of the main actors—people with different types of disabilities—to be heard. This is a very diverse group, and learning about their characteristics can provide valuable information. As the results of our research have shown, the diversity of needs and expectations is influenced mainly by the type of disability but also by the country of residence. The tourism industry should develop solutions to prepare an offer tailored to specific needs. This can be performed by improving one’s knowledge about disability, which increases sensitivity and encourages action. As our research results have shown, the lack of preparation of service staff is one of the most serious barriers to travel and touristic activity for people with disabilities. Therefore, it is worth ensuring proper training to the staff about different types of disabilities. It is also worthwhile to continue primary research among people with different types of disabilities using external funding opportunities, also in areas indirectly related to research, such as the European Social Fund and the Erasmus + Programme.

## Figures and Tables

**Table 1 ijerph-19-10575-t001:** A research sample—structure according to selected characteristics (*N* = 131).

Characteristic	Characteristic Categories	Percentage of Respondents
Country	Cyprus	24
	Greece	23
	Poland	30
	Portugal	23
Gender	Female	52
	Male	46
	Prefer not to say	2
Age	Up to 25 years old	6
	26–35 years old	28
	36–45 years old	32
	46–55 years old	14
	56–65 years old	10
	65+ years old	6
	Prefer not to say	4
Level of education	Primary	3
	Vocational	23
	Secondary	38
	Higher	32
	Prefer not to say	4
Place of residence	Village	5
	Town with up to 50,000 residents	5
	City with 50,000–100,000 residents	15
	City with 101,000–500,000 residents	41
	City with more than 500,000 residents	33
	Prefer not to say	1

**Table 2 ijerph-19-10575-t002:** Most popular companion during travel when taking trips with at least one overnight stay.

Companion	Percentage
Friends	35.1
Spouse/partner	32.8
Family	29.8
Alone	14.5
Travel assistant	2.3
Strangers	1.5

**Table 3 ijerph-19-10575-t003:** The main obstacles or barriers faced when travelling.

Obstacle/Barrier	Percentage
Insufficient financial resources	50.4
Lack of adequate information about attractions, e.g., lack of information in Braille, lack of translation into Sign Language, lack of QR codes in tourist places, inaccessible websites, etc.	47.3
Lack of attractions, e.g., no attractions or facilities dedicated to people with disabilities available at travel agencies	36.6
Lack of well-trained staff (unable to provide services to people with disabilities)	33.6
Health state/disability	22.1
Barriers in the means of transport	22.1
Lack of knowledge of foreign languages	22.1
Architectural barriers in or around tourist attractions (no ramps, steep roads, etc.)	18.3
Architectural barriers in accommodation (no lift, winding stairs, etc.)	17.6
Lack of travel companion	13.7
Other	6.9
No barriers to travelling	6.1
Lack of support/acceptance of trips from family or friends	3.8

**Table 4 ijerph-19-10575-t004:** Most important obstacles or barriers by country.

Rank	Country			
	Cyprus	Greece	Poland	Portugal
1	Lack of adequate information about the attractions; Lack of attractions	Insufficient financial resources	Insufficient financial resources	Lack of adequate information about the attractions
2	Insufficient financial resources	Lack of well-trained staff	Lack of adequate information about the attractions; Lack of attractions	Insufficient financial resources
3	Lack of well-trained staff	Lack of attractions	Lack of well-trained staff	Lack of well-trained staff; Health state/disability; Lack of attractions
4	Health state/disability	Barriers in the means of transport	Architectural barriers in accommodation	Lack of knowledge of foreign languages
5	Lack of knowledge of foreign languages	Lack of adequate information about attractions; Architectural barriers in or around tourist attractions	Barriers in the means of transport	Lack of travel companion

**Table 5 ijerph-19-10575-t005:** Factors that would make travelling easier.

Factor	Percentage
Offers from travel agencies dedicated to people with disabilities	58.8
More financial resources (own)	51.1
Information on adjusting the attractions to the needs of people with disabilities	49.6
Co-financing of trips by public institutions/non-governmental organizations	45.8
Well-trained staff (trained personnel to provide services to people with special needs, including those with disabilities)	45.0
Assistant for people with disabilities (at railway stations, airports, museums, etc.)	38.9
Better accessibility of tourist attraction websites to people with disabilities	38.2
A travel companion	27.5
Removing barriers from the means of transport and tourist attractions (churches, monuments, museums, open-air museums)	22.9
Availability of medical facilities at the destination point	21.4
Removing barriers in accommodation (lifts, adjusting bathrooms, etc.)	18.3
Personal assistants for people with disabilities who could accompany them during the journey	16.8
Other	8.4
Greater acceptance of trips on the part of family/friends	6.9

**Table 6 ijerph-19-10575-t006:** Most important factors that would make travelling easier by country.

Rank	Country			
	Cyprus	Greece	Poland	Portugal
1	Trips co-financing	Offers for PwD	Offers for PwD	Trips co-financing
2	Assistant for people with disabilities; Offers for PwD	Information about attractions for PwD; Well-trained staff	More financial resources (own)	Information about attractions for PwD
3	Information about attractions for PwD; Accessible tourism websites	More financial resources (own)	Information about attractions for PwD	More financial resources (own)
4	More financial resources (own)	Assistant for people with disabilities	Well-trained staff; Trips co-financing	Well-trained staff
5	Well-trained staff	Accessible tourism websites	Accessible tourism websites	Assistant for people with disabilities

**Table 7 ijerph-19-10575-t007:** Importance of various aspects of the trips.

Aspect	Average
Accessibility of facilities for people with disabilities	3.16
Tasting regional dishes and beverages	3.10
Visiting monuments, works of art, theatres, etc.	3.00
Experiencing an adventure	2.79
Meeting new people	2.73
Improving health and well-being	2.73
Rest and relaxation	2.72
Visiting family and friends	2.58
Visiting craftsmen and learning about local handicrafts	2.28

**Table 8 ijerph-19-10575-t008:** Results of t-tests for the importance of aspects of the trips with respect to gender and type of sensory disability.

Aspect	Gender		Type of Sensory Disability	
	t Statistic	*p*-Value	t Statistic	*p*-Value
Accessibility of facilities for people with disabilities	0.305	0.761	1.500	0.137
Tasting regional dishes and beverages	0.920	0.359	3.174	0.002
Visiting monuments, works of art, theatres, etc.	2.503	0.014	−1.630	0.106
Experiencing an adventure	−1.082	0.281	2.695	0.008
Meeting new people	−1.442	0.152	1.140	0.257
Improving health and well-being	−0.746	0.457	3.474	0.001
Rest and relaxation	−1.443	0.152	3.025	0.003
Visiting family and friends	−0.176	0.861	0.051	0.959
Visiting craftsmen and learning about local handicrafts	−0.188	0.851	2.513	0.014

**Table 9 ijerph-19-10575-t009:** Results of ANOVA for the importance of aspects of the trips with respect to country.

Aspect	F Statistic	*p*-Value
Accessibility of facilities for people with disabilities	2.340	0.077
Tasting regional dishes and beverages	4.055	0.009
Visiting monuments, works of art, theatres, etc.	9.583	0.000
Experiencing an adventure	8.573	0.000
Meeting new people	3.448	0.019
Improving health and well-being	22.820	0.000
Rest and relaxation	7.717	0.000
Visiting family and friends	14.343	0.000
Visiting craftsmen and learning about local handicrafts	7.571	0.000

**Table 10 ijerph-19-10575-t010:** Aspects influencing the choice of accommodation.

Aspect	Average
Location/good public transport connection to the city centre	3.60
Additional services included in the price (e.g., free Wi-Fi, breakfast included in the price, etc.)	3.50
Price	3.46
Standard (quality of service)	3.21
Possibility of cancelling the reservation free-of-charge	3.15
Reviews and opinions about the place	3.11
Facilities for people with disabilities (ramps, induction loops, the staff using Sign Language, etc.)	3.09
Website dedicated to people with visual impairment	2.25

**Table 11 ijerph-19-10575-t011:** Results of t-tests for aspects influencing the choice of accommodation with respect to gender and type of sensory disability.

Aspect	Gender		Type of Sensory Disability	
	t Statistic	*p*-Value	t Statistic	*p*-Value
Location/good public transport connection to the city centre	0.228	0.820	1.240	0.218
Additional services included in the price (e.g., free Wi-Fi, breakfast included in the price, etc.)	−2.105	0.037	1.028	0.307
Price	−0.372	0.771	1.238	0.219
Standard (quality of service)	0.216	0.830	1.784	0.077
Possibility of cancelling the reservation free-of-charge	−0.462	0.645	2.586	0.011
Reviews and opinions about the place	0.463	0.644	−1.325	0.188
Facilities for people with disabilities (ramps, induction loops, the staff using Sign Language, etc.)	0.633	0.528	0.936	0.351
Website dedicated to people with visual impairment	0.648	0.518	6.735	0.000

**Table 12 ijerph-19-10575-t012:** Results of ANOVA for aspects influencing the choice of accommodation with respect to country.

Aspect	F Statistic	*p*-Value
Location/good public transport connection to the city centre	0.935	0.426
Additional services included in the price (e.g., free Wi-Fi, breakfast included in the price, etc.)	1.118	0.345
Price	4.897	0.003
Standard (quality of service)	5.234	0.002
Possibility of cancelling the reservation free-of-charge	3.801	0.012
Reviews and opinions about the place	0.837	0.476
Facilities for people with disabilities (ramps, induction loops, the staff using Sign Language, etc.)	0.774	0.511
Website dedicated to people with visual impairment	17.212	0.000

## Data Availability

The dataset presented in the study is available upon request from the corresponding author.

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
