# Peer review of "Travelling from Perspective of Persons with Disability: Results of an International Survey"

_ijerph, 2022, doi:10.3390/ijerph191710575_

Round 1

Reviewer 1 Report

I enjoyed reading this manuscript. The findings are interesting and insightful, though I found the manuscript more in line with the classification of “in progress” as I’d like to see some minor revisions to improve it before publication.

Human rights of people with special needs/disabilities are the state’s responsibility so long as the person is a citizen of that particular nation. On the other hand, international tourists are at the mercy of the destination state’s policies and actions of the private sector. Even if EU members must follow EU directives, unless they are legally binding, member states can find loopholes in order to not-completely-following-the-directives. As such, the conclusion and discussion must be more than stating the “tourism industry must improve accessibility for this niche market.”

The responses from the respondents are intriguing. However, if the authors could identify the needs of tourists with disabilities in terms of the private business level, community level, national standards/regulations level and tourist’s personal level, the findings would be more meaningful.

I am “assuming” the respondents are answering about their “international” travel experiences as well as their “domestic” travel experiences – is that correct? It is not really reader-friendly if the reader must make “assumtions” which is required in this case. Please clearly describe if respondents are answering about both travel experiences or not.

Where were the subjects surveyed? The nations of the subjects are indicated, yet, the locations where the individual survey were completed is not mentioned. If there were any sensory disabilities, how was the data collection assisted? I am “assuming” the respondents were in their home countries at the time of survey, but is this correct? If it was a pen-and-paper questionnaire, how many researchers, research assistants or caretakers were involved and were they equally trained in conducting the survey? In what language were respondents questioned? Is the survey focused on tourist experiences of past travels to a foreign/international destination?

In this manuscript, I do not see literature from tourism researchers on accessibility and disability when numerous tourism researchers have published on this topic. Even though the manuscript has not been submitted to a tourism and hospitality journal, the potential readers are unquestionably involved with tourism and hospitality. I suggest incorporating literature from tourism researchers. A quick Internet search with relevant key words will find a numerous studies written by tourism authors. Only if this manuscript builds upon the existing literature from different disciplines (i.e., tourism and hospitality research), will the findings and discussions will become more convincing.

Author Response

Dear Sir or Madam,

Thank you for all your comments and remarks. We carefully analyzed all of them. Below you will find our responses as well as suggested changes.

Human rights of people with special needs/disabilities are the state’s responsibility so long as the person is a citizen of that particular nation. On the other hand, international tourists are at the mercy of the destination state’s policies and actions of the private sector. Even if EU members must follow EU directives, unless they are legally binding, member states can find loopholes in order to not-completely-following-the-directives. As such, the conclusion and discussion must be more than stating the “tourism industry must improve accessibility for this niche market.”

We completely agree with the reviewer’s statement. In this article, we mainly strove to show that even the best law will not ensure openness and full inclusion of people with disability. The most important thing is people’s attitudes and openness. Identifying disability only with mobility impairments is erroneous and often results in a belief that a given facility or services are fully accessible only because there is a ramp or lift. Our respondents often mentioned a barrier associated with low awareness of different types of disability displayed by the service staff. Therefore, we believe that it is necessary to ensure some training in this area, which will undoubtedly be a step towards a more open tourism industry. Our subsequent research will provide answers to more specific questions.

The responses from the respondents are intriguing. However, if the authors could identify the needs of tourists with disabilities in terms of the private business level, community level, national standards/regulations level and tourist’s personal level, the findings would be more meaningful.

These goals are very ambitious and difficult to achieve. However, we have already obtained some financial resources for our next research on travel of people with disability. This time, we conducted an international comparative study among respondents belonging to the so-called “group with special needs”, which, in addition to people with disability, also includes elderly people, pregnant women and people with young children. We collected over 1,200 questionnaire forms and hope that the analysis of the results of this survey will provide answers to many interesting questions and issues related to travel.

I am “assuming” the respondents are answering about their “international” travel experiences as well as their “domestic” travel experiences – is that correct? It is not really reader-friendly if the reader must make “assumptions” which is required in this case. Please clearly describe if respondents are answering about both travel experiences or not.

The respondents answered questions about their travel experiences. It was about both international and domestic travel.

Where were the subjects surveyed? The nations of the subjects are indicated, yet, the locations where the individual survey were completed is not mentioned. If there were any sensory disabilities, how was the data collection assisted? I am “assuming” the respondents were in their home countries at the time of survey, but is this correct? If it was a pen-and-paper questionnaire, how many researchers, research assistants or caretakers were involved and were they equally trained in conducting the survey? In what language were respondents questioned? Is the survey focused on tourist experiences of past travels to a foreign/international destination?

The survey was translated into the national languages of all countries participating in the research. As far as Cyprus and Greece were concerned, we used the same translation. The survey was carried out by trained members of the project teams. As for deaf respondents using sign language, a sign language interpreter specific to the their country of residence was involved in completing the survey. The survey was prepared in a form accessible to visually impaired or blind people (audio description). The survey focused on respondents’ experiences connected with international or domestic travel. The main objective was to identify possible barriers and issues that make travelling difficult.

Due to doubts concerning the way the research was carried out, we suggest adding the following text in the “Materials and Methods” section, current line 97 and the subsequent ones (after reviews start from 107):

“The respondents answered questions about their experiences connected with both international and domestic travel. The survey questionnaires were translated into the national languages of all participating countries. As far as Cyprus and Greece were concerned, we used the same translation. The survey was carried out face to face by trained project team members in the respondents’ countries of residence. As for deaf respondents using sign language, a sign language interpreter specific to their country of residence was involved in completing the survey. The survey was prepared in a form accessible to visually impaired or blind people (audio description). The questionnaire was completed by the interviewer on the basis of the respondent’s statements, who also ultimately accepted the interview transcripts.”

In this manuscript, I do not see literature from tourism researchers on accessibility and disability when numerous tourism researchers have published on this topic. Even though the manuscript has not been submitted to a tourism and hospitality journal, the potential readers are unquestionably involved with tourism and hospitality. I suggest incorporating literature from tourism researchers. A quick Internet search with relevant key words will find a numerous studies written by tourism authors. Only if this manuscript builds upon the existing literature from different disciplines (i.e., tourism and hospitality research), will the findings and discussions will become more convincing.

The literature was supplemented with articles from the field of tourism and hospitality. We highlighted the relationship between inclusive tourism and social and economic development, and we also pointed to research on PwD tourism related, in particular, to transport, accommodation and accessibility of tourist attractions.

We suggest adding the following paragraph to the “Introduction” section from the current line 58:

“In the literature, attention is paid to the relationship between inclusive tourism and social and economic development [10], and the industry is believed to have the potential to provide additional social and economic benefits [11]. What is more, Liasidou et al. [12] emphasise that the issues can be analysed in a broader spectrum and concern other groups of people with special needs (e.g. elderly people, pregnant women or people travelling with young children). The research on accessible tourism concerns different aspects connected with travelling, some of which being related to transport [13,14,15,16,17]. Other research concerns the accessibility of accommodation [3,18,19,20,21] and different tourist attractions [4,5,22,23,24].”

Supplemented literature:

Scheyvens, R., & Biddulph, R. (2018). Inclusive tourism development. Tourism Geographies20(4), 589-609.

Biddulph, R., & Scheyvens, R. (2018). Introducing inclusive tourism. Tourism Geographies20(4), 583-588.

Liasidou, S., Umbelino, J., & Amorim, É. (2019). Revisiting tourism studies curriculum to highlight accessible and inclusive tourism. Journal of Teaching in Travel & Tourism19(2), 112-125.

Poria, Y., Reichel, A., & Brandt, Y. (2010). The flight experiences of people with disabilities: an exploratory study. Journal of Travel Research49(2), 216-227.

Lipp, E. (2015). What creates access and inclusion at airports? Journal of Airport Management9(4), 390-397.

Chang, Y. C., & Chen, C. F. (2012). Meeting the needs of disabled air passengers: Factors that facilitate help from airlines and airports. Tourism Management33(3), 529-536.

Sanmargaraja, S., & Wee, S. T. (2015). Accessible transportation system for the disabled tourist in the national park of Johor State, Malaysia. International Journal of Social Science and Humanity5(1), 15.

Darcy, S., & Pegg, S. (2011). Towards strategic intent: Perceptions of disability service provision amongst hotel accommodation managers. International Journal of Hospitality Management30(2), 468-476.

Kim, W. G., Stonesifer, H. W., & Han, J. S. (2012). Accommodating the needs of disabled hotel guests: Implications for guests and management. International Journal of Hospitality Management31(4), 1311-1317.

Tutuncu, O. (2017). Investigating the accessibility factors affecting hotel satisfaction of people with physical disabilities. International Journal of Hospitality Management65, 29-36.

Poria, Y., Reichel, A., & Brandt, Y. (2011). Dimensions of hotel experience of people with disabilities: an exploratory study. International Journal of Contemporary Hospitality Management23(5), 571-591.

Israeli, A. A. (2002). A preliminary investigation of the importance of site accessibility factors for disabled tourists. Journal of Travel Research41(1), 101-104.

Jamaludin, M., & Kadir, S. A. (2012). Accessibility in Buildings of Tourist Attraction: A case studies comparison. Procedia-Social and Behavioral Sciences35, 97-104.

Santana-Santana, S. B., Peña-Alonso, C., & Espino, E. P. C. (2020). Assessing physical accessibility conditions to tourist attractions. The case of Maspalomas Costa Canaria urban area (Gran Canaria, Spain). Applied Geography125, 102327.

Reviewer 2 Report

The topic is interesting and timely relevant in terms of inclusiveness in the tourism sector. I recommend author(s) investigate more the domain of inclusive growth, dealing with traveling with disabled people.  The paper is simple and clear, but significant for the future tourism industry. 

This paper is timely and relevant for extending the topic in the tourism literature. The findings are educational and informative to understanding barriers for disabled people to travel in cross-cultural settings. At that point, I judged this paper "accepted with minor revision." However, this paper has serious flaws in not presenting the existing literature review regarding traveling by disabled people. I would like to ask authors to supplement these literature reviews extensively. Also, the paper has a relatively small size of samples (131). This paper should have done both reliability and validity tests. The results with 131 samples cannot be generalized. Nevertheless, the findings of this paper are significant to my understanding of inclusive tourism phenomena (Tourism for all).  

Author Response

Dear Sir or Madam,

Thank you for all your comments and remarks. We carefully analyzed all of them. Below you will find our responses as well as suggested changes.

The topic is interesting and timely relevant in terms of inclusiveness in the tourism sector. I recommend author(s) investigate more the domain of inclusive growth, dealing with traveling with disabled people.  The paper is simple and clear, but significant for the future tourism industry.

We would like to inform you that we have already obtained some financial resources for our next research on travel of people with disability. This time, we conducted an international comparative study among respondents belonging to the so-called “group with special needs”, which, in addition to people with disability, also includes elderly people, pregnant women and people with young children. We collected over 1,200 questionnaire forms and hope that the analysis of the results of this survey will provide answers to many interesting questions and issues related to travel. The results of the conducted analyses will be presented in subsequent articles.

This paper is timely and relevant for extending the topic in the tourism literature. The findings are educational and informative to understanding barriers for disabled people to travel in cross-cultural settings. At that point, I judged this paper "accepted with minor revision." However, this paper has serious flaws in not presenting the existing literature review regarding traveling by disabled people. I would like to ask authors to supplement these literature reviews extensively. Also, the paper has a relatively small size of samples (131). This paper should have done both reliability and validity tests. The results with 131 samples cannot be generalized. Nevertheless, the findings of this paper are significant to my understanding of inclusive tourism phenomena (Tourism for all). 

We are aware of the limitations resulting from the sample size. We never intended to generalise the obtained results. As we indicated earlier, we have carried out further research on a much larger population, which will undoubtedly increase the statistical value of the results we achieved. Concerning the literature, we would like to inform you that it has been supplemented with articles from the field of tourism and hospitality. We highlighted the relationship between inclusive tourism and social and economic development, and we also pointed to research on PwD tourism related, in particular, to transport, accommodation and accessibility of tourist attractions.

We suggest adding the following paragraph to the “Introduction” section from the current line 58:

“In the literature, attention is paid to the relationship between inclusive tourism and social and economic development [10], and the industry is believed to have the potential to provide additional social and economic benefits [11]. What is more, Liasidou et al. [12] emphasise that the issues can be analysed in a broader spectrum and concern other groups of people with special needs (e.g. elderly people, pregnant women or people travelling with young children). The research on accessible tourism concerns different aspects connected with travelling, some of which being related to transport [13,14,15,16,17]. Other research concerns the accessibility of accommodation [3,18,19,20,21] and different tourist attractions [4,5,22,23,24].”

Supplemented literature:

Scheyvens, R., & Biddulph, R. (2018). Inclusive tourism development. Tourism Geographies20(4), 589-609.

Biddulph, R., & Scheyvens, R. (2018). Introducing inclusive tourism. Tourism Geographies20(4), 583-588.

Liasidou, S., Umbelino, J., & Amorim, É. (2019). Revisiting tourism studies curriculum to highlight accessible and inclusive tourism. Journal of Teaching in Travel & Tourism19(2), 112-125.

Poria, Y., Reichel, A., & Brandt, Y. (2010). The flight experiences of people with disabilities: an exploratory study. Journal of Travel Research49(2), 216-227.

Lipp, E. (2015). What creates access and inclusion at airports? Journal of Airport Management9(4), 390-397.

Chang, Y. C., & Chen, C. F. (2012). Meeting the needs of disabled air passengers: Factors that facilitate help from airlines and airports. Tourism Management33(3), 529-536.

Sanmargaraja, S., & Wee, S. T. (2015). Accessible transportation system for the disabled tourist in the national park of Johor State, Malaysia. International Journal of Social Science and Humanity5(1), 15.

Darcy, S., & Pegg, S. (2011). Towards strategic intent: Perceptions of disability service provision amongst hotel accommodation managers. International Journal of Hospitality Management30(2), 468-476.

Kim, W. G., Stonesifer, H. W., & Han, J. S. (2012). Accommodating the needs of disabled hotel guests: Implications for guests and management. International Journal of Hospitality Management31(4), 1311-1317.

Tutuncu, O. (2017). Investigating the accessibility factors affecting hotel satisfaction of people with physical disabilities. International Journal of Hospitality Management65, 29-36.

Poria, Y., Reichel, A., & Brandt, Y. (2011). Dimensions of hotel experience of people with disabilities: an exploratory study. International Journal of Contemporary Hospitality Management23(5), 571-591.

Israeli, A. A. (2002). A preliminary investigation of the importance of site accessibility factors for disabled tourists. Journal of Travel Research41(1), 101-104.

Jamaludin, M., & Kadir, S. A. (2012). Accessibility in Buildings of Tourist Attraction: A case studies comparison. Procedia-Social and Behavioral Sciences35, 97-104.

Santana-Santana, S. B., Peña-Alonso, C., & Espino, E. P. C. (2020). Assessing physical accessibility conditions to tourist attractions. The case of Maspalomas Costa Canaria urban area (Gran Canaria, Spain). Applied Geography125, 102327.